# Effect of MMP/TIMP Balancing of *Cynoglossus semilaevis* Shell Extracts on Skin Protection

**Soo-Cheol Choi and In-Ah Lee \***

Department of Chemistry, College of Natural Science, Kunsan National University, 558 Daehak-ro, Gunsan-si 54150, Jeollabuk-do, Korea; csc2714@hanmail.net
* Correspondence: leeinah@kunsan.ac.kr; Tel.: +82-063-469-4574

**Abstract:** *Cynoglossus semilaevis* shell is a by-product of the *Cynoglossus semilaevis*, a species of fish mainly distributed along the west coast of Korea. As its skin is very tough and difficult to process, it is not useful as food. For this reason, most of it is discarded except for a small amount that is used as feed, which results in environmental pollution. Considering this, there is a need for research on the development of functional materials using *Cynoglossus semilaevis* shell. This study focused on the mechanism of in vitro expression function of *Cynoglossus semilaevis* shell extract (CSE) for skin tissue in human dermal fibroblasts that induced or did not induce wrinkles by UV-B irradiation and aims to use it as a functional material for human skin beauty or wrinkle improvement through extraction and purification. According to the ELISA results using human dermal fibroblast cells, CSE reduced MMP-1 and elastase activity by up to 21.89% and 12.04%, respectively, in a concentration-dependent manner, and increased PIP synthesis by up to 62.24% in a concentration-dependent manner. The RT-PCR test results using mRNA showed the MMP-1, 2, and 3 expression levels were suppressed in the CSE-treated group compared to the UVB-induced group and caused a concentration-dependent increase in TIMP-1 in the CSE-treat group. These results suggest that CSE can maintain and improve skin tissue conditions through MMP/TIMP balancing in human dermal fibroblast cell lines and indicate its potential as a functional material for improving skin diseases and suppressing photo-aging.

**Keywords:** *Cynoglossue semilaevis* shell; MMPs; TIMP-1; collagen; anti-aging

## 1. Introduction

Wrinkles are caused by a decrease in hormone levels with age, exposure to UV rays, or physical causes such as repeated muscle use and facial expressions and are characterized by the thinning of the dermis, causing epidermis thinning due to a decrease in the proliferation rate of cells that occupy the basal cell layer of the epidermis [1,2].

The dermis is a thick layer of skin beneath the epidermis, which makes up about 90% of our skin. It mainly consists of connective and elastic fibers composed of hyaluronic acid, collagen, and elastin, which improve skin elasticity [3–5]. The collagen and elastin in the dermal layer are strong connective proteins that form the dermal matrix, which prevents our skin from being easily damaged by external shocks and is known to play a significant role in maintaining skin elasticity. As such, the decomposition of collagen and elastin in human connective tissue directly affects skin elasticity and wrinkle formation [6,7].

MMPs are transcriptionally regulated by cell growth factors, hormones, and cytokines and are essential enzymes for biological processes such as blastogenesis, organogenesis, nerve growth, ovulation, angiogenesis, and tissue absorption and remodeling by degrading ECM [8–12]. However, an imbalance with TIMPs (endogenous inhibitors), which can be caused by various in vivo factors, is known to lead to excessive increases in MMPs, which can cause inflammatory skin diseases due to skin tissue damage [13].

Of the various types of MMPs, MMP-1 is a proteolytic enzyme specific to collagen. Although the synthesis of type 1 collagen and MMP-1 (degrading enzyme) is balanced in

normal skin, imbalances between type 1 collagen and MMP-1 appear in aging skin due to internal and external factors [8,14–16].

As major inhibitors of MMP, four homologous TIMPs have been identified thus far [17]. Among them, TIMP-1 is an inhibitory molecule that regulates the expression of various MMPs in the skin and a cytoprotective factor that inhibits the breakdown of intercellular substances found in aging and cell damage caused by aging and exogenous stimuli and is known to promote cell differentiation and have anti-apoptotic effects [18–20]. TIMP-1 is a natural inhibitor of a wide range of metal-dependent proteolytic enzymes present in almost all living tissues, including human skin, and is known as a biomolecule that inhibits ECM degradation, such as collagen, elastin, gelatin, fibronectin, and keratin by proteolytic enzymes most prominent in natural and extrinsic aging [21–24].

The tongue sole (*Cynoglossus semilaevis*) is a benthic fish belonging to the *Cynoglossidae* family in Pleuronectiformes, typically found on the muddy bottoms along the coast or in brackish or freshwater areas. Tongue sole, commonly known as the tonguefish, are caught around the west coast of Korea, such as Gunsan and Seocheon. It is the largest species in the *Cynoglossidae* family and is very useful from an industrial perspective because of its rapid growth rate [25,26]. Tongue sole skin is a by-product of tongue soles living in the west coast. Although tongue soles are effective for bone health and preventing osteoporosis due to their high content of vitamins, folic acid, potassium, and calcium, their skin is tough and difficult to process, which means it is not useful as food. For this reason, most of it is discarded, except for a small amount used as feed, which results in environmental pollution. Most fish skins are composed almost solely of collagen, thus the collagen extracted and refined from fish skin can be used for food and medicine (Figure 1). This study proposes the use of tongue sole skin to develop functional materials as a therapeutic agent for skincare or treating skin diseases.

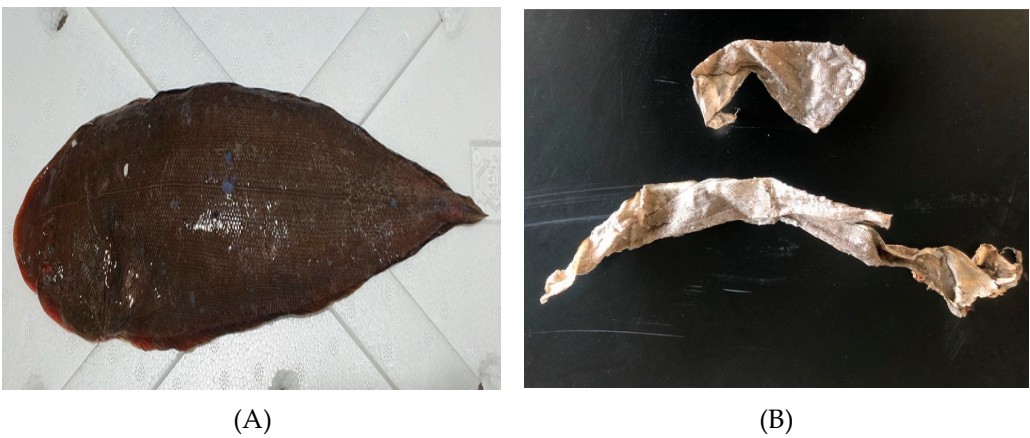

(A)　　　　　　　　　　　　　　　　　　　　　　　　　(B)

**Figure 1.** *Cynoglossus semilaevis* (**A**), a by-product of *Cynoglossus semilaevis* shell (**B**).

## 2. Reagents and Methods

### 2.1. Materials and Reagents

Dulbecco's Modified Eagle's Medium/F12 Nutrient Mixture Ham (DMEM/F12 3:1) (Gibco, Waltham, MA, USA), Anti-biotic Antimycotic (Gibco, Waltham, MA, USA), Trypsin-EDTA (Gibco, Waltham, MA, USA), Phosphate Buffered Saline (Gibco, Waltham, MA, USA), Fetal Bovine Serum (FBS) (Gibco, Waltham, MA, USA), 3-(4,5-Dimethylthiazol-2-yl)-2,5-Diphenyl-2H-tetrazolium Bromide (MTT) (Sigma-Aldrich, St Louis, MO, USA), TGF-β1 (Sigma-Aldrich, St Louis, MO, USA), Albumin from Bovine Serum (BSA) (Sigma-Aldrich, St Louis, MO, USA), Bradford Reagent (BIORAD, Hercules, CA, USA), 1 M Tris-HCl pH 8.0 (Elpis-biotech, Daejeon, Korea), Triton X-100 (Yakuri Pure Chemicals, Kyoto, Japan), Dimethyl Sulfoxide (DMSO) (Amresco, Solon, OH, USA), Pro-collagen Type I Peptide (PIP) ELISA Kit (Takara, Shiga, Japan), Matrix Metalloproteinase-1 (MMP-1) Human ELISA kit

(Abcam, Cambridge, MA, USA), Phosphoramidon (Sigma-Aldrich, St Louis, MO, USA), STANA (Sigma-Aldrich, St Louis, MO, USA)

### 2.2. Preparation of Cynoglossus semilaevis Shell Extract (CSE)

After washing the *Cynoglossus semilaevis* shell, it was naturally dried and finely pulverized to prepare *Cynoglossus semilaevis* shell powder. A 10-fold weight of 60% ethanol aqueous solution was added to the *Cynoglossus semilaevis* peel powder, extracted for 12 h at room temperature, and then filtered through Whatman filter paper to prepare a *Cynoglossus semilaevis* peel extract. The extracted CSE was stored in a $-80\,^{\circ}$C freezer for 1 day and then powdered using a rotary vacuum concentrator, and the powdered *Cynoglossus semilaevis* peel extract was diluted in DMSO at a concentration of 50 mg/mL and stored at $-20\,^{\circ}$C for use in experiments.

### 2.3. Amino Acid Composition Analysis

To analyze the amino acid composition contained in CSE, 100 mg of a CSE sample was put in a glass tube, dissolved in 3 mL of 6 N-HCl solution, and acid hydrolyzed in a hot air dryer at 110 $^{\circ}$C for 24 h. Thereafter, the CSE sample was derivatized with PITC (Phenyl isothiocyanate) and used as a sample for analysis. The amino acid composition of the sample for analysis was analyzed using HPLC (HPLC PI-CO-TAG system). The HPLC used for the analysis was a Water HPLC System (Waters 510 HPLC Pump), Waters 717 automatic sampler, Waters 996 photodiode array detector, Waters gradient controller, Millenium 2010 chromatography manager (Waters Co.), and the column was a free amino acid analysis column (3.9 mm $\times$ 300 mm, 4 m waters pi-co-Tag, Pico Rivera, CA, USA). The column temperature was 40 $^{\circ}$C, and the flow rate was 1 mL/min. As the mobile phase, 140 $\mu$M sodium acetate (6% acetonitrile) was used for buffer A, and 60% acetonitrile was used for buffer B.

### 2.4. Cell Line Selection and Cell Culture

Human Dermal Fibroblasts (HDFs) cells were selected and used based on the efficacy test data of ⌜Guideline for Efficacy Evaluation of Functional Cosmetics (II)⌟. HDFs (ATCC, Manassas, VA, USA) were primary cells isolated from adult skin and were used in which Mycoplasma, Hepatitis B, Hepatitis C, HIV-1, Bacteria, Yeast, and other fungi were not detected. HDFs were inoculated on the bottom of a culture dish and cultured in a 37 $^{\circ}$C, 5% $CO_2$ incubator (HERAcell 150i, Thermo, Waltham, MA, USA) by adding DMEM/F12 3:1 mixed medium containing 1% Antibiotic-Antimycotic and 10% FBS.

### 2.5. MTT Assay

Single HDFs cells suspensions with a density of $1 \times 10^6$ cells/well were plated to a 24-well plate and cultured overnight. Then, the cells were treated with 1.56, 3.13, 6.25, 12.5, 25, and 50 mg/mL CSE. After 24 h, 10 $\mu$L MTT stock (5 mg/mL) was added to each well and incubated for 2 h. Then, the supernatants were discarded, and 100 $\mu$L DMSO was added to each well. After 10 min of shaking, the optical density at the wavelength of 570 nm (OD570) was measured on a SPARK microplate reader (Bio Tek, Winooski, VT, USA). All samples were tested in triplicate and repeated 3 times [27–29].

### 2.6. UV-B Irradiation

HDFs were irradiated by UV-B lamp GL20SE (Sankyo denki, Japan) fitted with a UV-B source designed for microplates. HDFs grown in microplates were irradiated at 20 mJ/cm$^2$ UV-B dose. Cells were irradiated in phosphate-buffered saline (PBS) without the plastic lid. When the irradiation received matched the desired programmed energy, the UV-B irradiation stopped automatically, and subsequently, the cells were incubated with DMEM without FBS until analysis.

### 2.7. Enzyme-Linked Immunosorbent Assay

Release of MMP-1 and type I pro-collagen in HDFs were analyzed by ELISA. Cells ($1 \times 106$ cells/well) were pre-incubated in 24-well plates for 24 h to reach >90% confluence and washed with PBS. The cells were treated with or without different concentrations (0.2, 0.39, 0.78, 1.56, 3.13, and 6.25 mg/mL) of CSE for 24 h. Cell culture medium from each well was analyzed for its MMP-1 (Abcam, USA) and type I pro-collagen (Takara, Japan) contents per the manufacturer's instructions of the ELISA kit.

### 2.8. Measurement of the Ability to Inhibit Elastase Activity

HDFs were inoculated and cultured in a 100 mm culture dish at $5 \times 10^4$ cells/well, and the cultured cells were dissolved by adding 0.1% Triton X-100 0.2 M Tris-HCl (pH 8.0) solution, and this cell lysate was dissolved at 4 °C. After centrifugation at 10,000 rpm for 10 min, the supernatant was taken, and an enzyme solution containing elastase was used. The elastase enzyme solution was quantified, 100 µg each was added to a 96-well plate, and 0.2 M Tris-HCl (pH 8.0) buffer was added to make 80 µL. The test substance was diluted in buffer and added by 10 µL, and 10 µL of STANA, a substrate of elastase, was added to each well and reacted for 90 min in an oven at 37 °C. The absorbance was measured at 405 nm to evaluate the elastase activity. Phosphoramidon (final treatment concentration 5.9 µg/mL) was used as a positive control.

### 2.9. Reverse Transcription Polymerase Chain Reaction (RT-PCR) Analysis

Cells ($1 \times 10^6$ cell/well) were grown to >90% confluence prior to UV-B irradiation. Following UV-B irradiation, cells were treated with or without CSE for 24 h. Total RNA was isolated from nonirradiated and irradiated (UV-B, 20 mJ/cm$^2$) HDFs using AccuPrep® Universal RNA Extraction Kit (Bioneer, Daejeon, Korea) according to manufacturer's instructions. Total RNA was treated with RNase-free DNase I (Thermo Fisher Scientific, Rockford, IL, USA). The cDNA synthesis from total RNA was performed using Cell Script All-in-One cDNA synthesis Master Mix (CellSafe, Yongin, Korea) following the producer's directions. cDNA amplification was carried out with quantitative PCR in a Thermal Cycler Dice® Real-Time System TP800 (Takara Bio Inc., Ohtsu, Japan) following the manufacturer's protocol. Briefly 20 µL of cDNA sample mixed with forward and reverse primers in nuclease-free water. The target cDNA was amplified using following forward and reverse primers; TNF-α, forward 5′-GAG CTG AGA GAT AAC CAG CTG GTG-3′, reverse 5′-CAG ATA GAT GGG CTC ATA CCA GGG-3′, IL-6, forward 5′-ATG AAC TCC TTC TCC ACA AGC-3′, reverse 5′-GTT TTC TGC CAG TGC CTC TTT G-3′, MMP-1, forward 5′-TGA AAA GCG GAG AAA TAG TGG-3′, reverse 5′-GAG GAC AAA CTG AGC CAC ATC-3′; MMP-2, forward 5′-TTG CCA TCC TTC TCA AAG TTG TAG G-3′, reverse 5′-CAC TGT CCA CCC CTC AGA GC-3′; MMP-3, forward 5′-CAC TCA CAG ACC TGA CTC GGT T-3′, reverse 5′-AAG CAG GAT CAC AGT TGG CTG G-3′; TIMP-1, forward 5′-ATA CTT CCA CAG GTC CCA CAA C-3′, reverse 5′-GGA TGG ATA AAC AGG GAA ACA C-3′; β-actin, forward 5′-AGA TCA GAT CAT TGC TCC TCC TG-3′, reverse 5′-CAA GAA AGG GTG TAA CGC AAC TAA G-3′. The PCR amplification was carried out with an initial denaturation at 95 °C for 1 min, followed by 40 PCR cycles, each cycle consisting of 95 °C for 15 s and 60 °C for 30 s. β-Actin was used as an internal control [30–33]. PCR results were isolated and quantified by beta-actin for quantitative comparison (FluoroBOX, Cellgentek, Korea).

### 2.10. Statistical Analysis

All data were expressed as mean value ± standard deviation, and statistical analysis was performed using SPSS® Package Pro-gram (IBM, Armonk, NY, USA). All results were the results of 3 or more independent experiments, and statistical significance was verified at the independent samples t-test $p < 0.05$ significance level.

## 3. Results

### 3.1. Amino Acid Composition Analysis

Table 1 shows the results of analyzing the amino acids and free amino acids of CSE by HPLC. Unlike other proteins, collagen contains hydroxylated amino acids hydroxyproline and hydroxylysine and is a three-dimensional macromolecular form in which three polypeptide chains are twisted together in a helical structure. The triple helix region of collagen is composed of a repeating amino acid sequence of Gly-XY, and proline and hydroxyproline are frequently located at the positions of X and Y. The amino acid contents of glycine, proline, and hydroxyproline confirm the presence or absence of collagen in the *Cynoglossus semilaevis* shell extract. In Table 1, CSE contains 17 amino acids such as proline, and glycine accounts for about 20% of the total amino acids as 3.905 μg/mL, and proline accounts for about 7.4% as 1.432 μg/mL. In addition, when checking the free amino acid content of CSE, it contains 20 kinds of free amino acids, including hydroxyl-L-proline, and it was detected high content that occupied more than 85% of total free amino acid. These results suggest that glycine, proline, and hydroxyproline, which constitute the repetitive Gly-X-Y amino acid sequence in the triple helix region of collagen, are contained in a large amount in CSE, and the collagen component contained in a large amount was estimated to have a positive effect on the skin (please see the attachment).

**Table 1.** Amino acid composition in *Cynoglossus semilaevis* shell extract (please see the attachment).

| *Cynoglossus semilaevis* **Shell Extract** | | | |
|---|---|---|---|
| **Amino Acid** | | **Physiological (Free) Amino Acid** | |
| **Kind** | **Content (μg/mL)** | **Kind** | **Content (μg/mL)** |
| Aspartic acid (Asp) | 1217 | o-Phosphoserine | 4 |
| Threonine (Thr) | 447 | Taurine | 33 |
| Serine (Ser) | 807 | o-Phosphoethanolamine | 1 |
| Glutamic acid (Glu) | 1980 | Sarcosine | 0 |
| Glycine (Gly) | 3905 | L-2-Aminoadipic Acid | 2 |
| Cysteine (Cys) | 4537 | L-Citruline | 8 |
| Alanine (Als) | 30 | DL-2-Aminobutyric | 0 |
| Valine (Val) | 582 | LCystathionine | 10 |
| Methionine (Met) | 417 | β-Alanine | 8 |
| Isoleucine (Ile) | 225 | DL-3-Aminoisobutyric Acid | 7 |
| Leucine (Leu) | 621 | 4-Aminobutyric acid | 1 |
| Tyrosine (Tyr) | 177 | 2-Aminoethanol | 133 |
| Phenylalanine (Phe) | 398 | DL-plusallo-δ-Hydroxylysine | 0 |
| Lysine (Lys) | 709 | L-Ornithine | 3 |
| Histidine (His) | 235 | L-1-Methylhistidine | 0 |
| Arginine (Arg) | 1562 | L-3-Methylhistidine | 0 |
| Proline (Pro) | 1432 | L-Anserine | 0 |
| | | L-Carnosine | 0 |
| | | hydroxyl-L-proline | 1554 |
| Total | 19,281 | Total | 1764 |

### 3.2. Effect of CSE on HDF Cells Viability

In order to assess the cytotoxic effect of CSE on HDF cells, an MTT assay was carried out. Various concentrations of CSE, including 1.56, 3.13, 6.25, 12.5, 25, and 50 mg/mL, were chosen to treat HDF cells for 24 h. As shown in Figure 2, compared to control cells treated with 0.00001 mg/mL TGF-β1, the cell viabilities of the following treatments with 25 and 50 mg/mL CSE were 45% and 50%, respectively, which indicated that HDF cells viability was not affected significantly below 12.5 mg/mL CSE for 24 h. Therefore, the concentration range of CSE for subsequent wrinkling improvement activity screening was determined to be within 12.5 mg/mL (Figure 2).

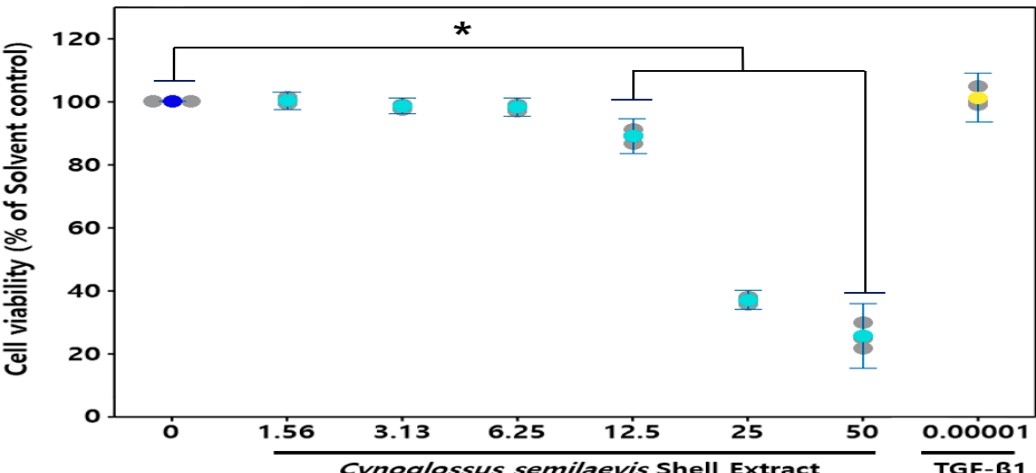

**Figure 2.** Cell viability in human dermal fibroblasts. The data are expressed as % of solvent control (Mean ± SD (n = 3), * $p < 0.05$, *Cynoglossus semilaevis* shell extract vs. solvent control). ●: Measures in each group, ●: Mean value of untreated group, ●: Mean value of CSE treatment group, ●: Mean value of positive control.

### 3.3. Effect of CSE on the Type I Pro-Collagen and MMP-1

CSE treatment at various concentrations (0.2, 0.39, 0.78, 1.56, 3.13, and 6.25 mg/mL) in HDF cells showed increased synthesis of type I pro-collagen compared to untreated groups. The basic type I pro-collagen level of HDF culture was 132.32 ng/mL, and the presence of CSE increased the type I pro-collagen synthesis depending on the concentration. Type I pro-collagen levels increased significantly from 29.69% to 62.24% concentration-dependent compared to solvent control after treatment of CSE (0.20 to 6.25 mg/mL) in HDF cells.

CSE-exposed cells released substantially decreased levels of MMP-1 compared to the non-exposed group (Figure 3B). The base MMP-1 level in the HDF culture medium was 4.3 ng/mL, and the presence of CSE suppressed the increase in MMP-1 levels in a concentration-dependent manner. The MMP-1 levels significantly reduced the concentration-dependent collagenase activity in cells between 12.41~21.89% compared to solvent controls after CSE (0.39~6.25 mg/mL) treatment.

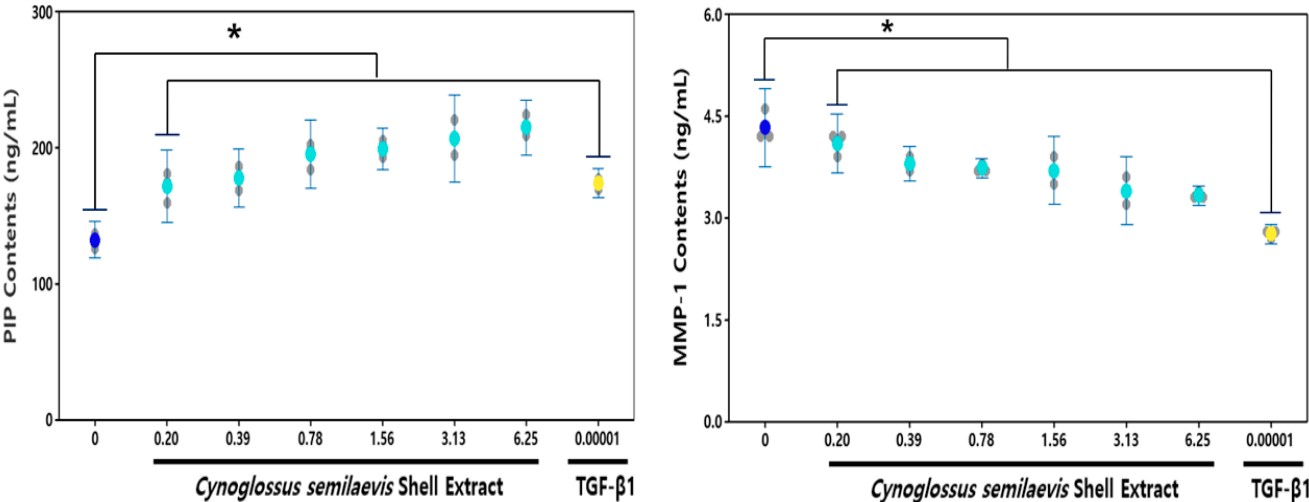

**Figure 3.** Effect of CSE on the cellular release of type Iα1 pro-collagen (**A**) and MMP-1 (**B**) in HDFs. The contents of type Iα1 pro-collagen and MMP-1 were calculated from HDF culture media following 24 h incubation with different concentrations of CSE using ELISA. The data are expressed as % of Solvent control (Mean ± SD (*n* = 3). * $p < 0.05$, *Cynoglossus semilaevis* shell extract vs. Solvent control). ●: Measures in each group, ●: Mean value of untreated group, ●: Mean value of CSE treatment group, ●: Mean value of positive control.

### 3.4. Effect of CSE on the Elastase

The treatment of various concentrations (0.2, 0.39, 0.78, 1.56, 3.13, and 6.25 mg/mL) of CSE in HDF cells reduced elastase levels compared to the group that did not treat CSE (Figure 4). Based on 100% of the basic elastase activities of HDF culture medium, treatment of CSE has concentration-dependent inhibition of an increase in elastase levels. Elastase levels significantly reduced concentration-dependent elastase activity in cells by 12.41% to 21.89%, compared to solvent controls after CSE (0.39 to 6.25 mg/mL) treatment.

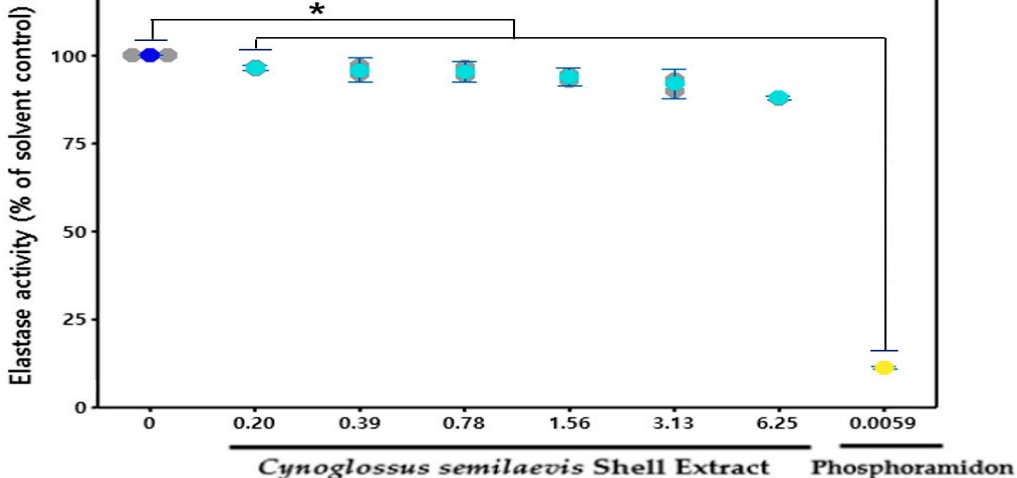

**Figure 4.** Elastase activity in a cell-free system. The data are expressed as % of solvent control (mean $\pm$ SD (n = 3). * $p < 0.05$, *Cynoglossus semilaevis* shell extract vs. solvent control). ●: Measures in each group, ●: Mean value of untreated group, ●: Mean value of CSE treatment group, ●: Mean value of positive control.

### 3.5. Effect of CSE on the UV-B Mediated Expression

UV-B-irradiated HDF cells were treated with CSE at different concentrations to analyze mRNA expression levels related to inflammation and photo-aging. In the UV-B-treated HDF cells, inflammation-related genes of TNF-alpha and IL-6 were strongly increased, and it was confirmed that the concentration-dependent reduction of inflammation when CSE was treated by concentration (Figure 5A).

Furthermore, in photon-aging-related gene expressions such as MMP-1,2 and TIMP-1 in HDF cells where UV-B was investigated, TIMP-1 and COL1A1 mRNA expressions were reduced, and MMP-1 and MMP-2 mRNA expressions were strongly increased. UV-B induced changes in the MMP-1,2 and TIMP-1 were ameliorated by CSE (1, 5, and 10 mg/mL) treatment in a concentration-dependent manner. In HDF cells treated with CSE, the mRNA expression of MMP-1 and 2 increased by UV-B irradiation was significantly reduced, and the mRNA expression of TIMP-1 decreased by UV-B irradiation was significantly increased. As the above result, the amount of type I$\alpha$1 pro-collagen biosynthesis was increased through the control of MMP/TIMP by treatment by the concentration of CSE (Figure 5B).

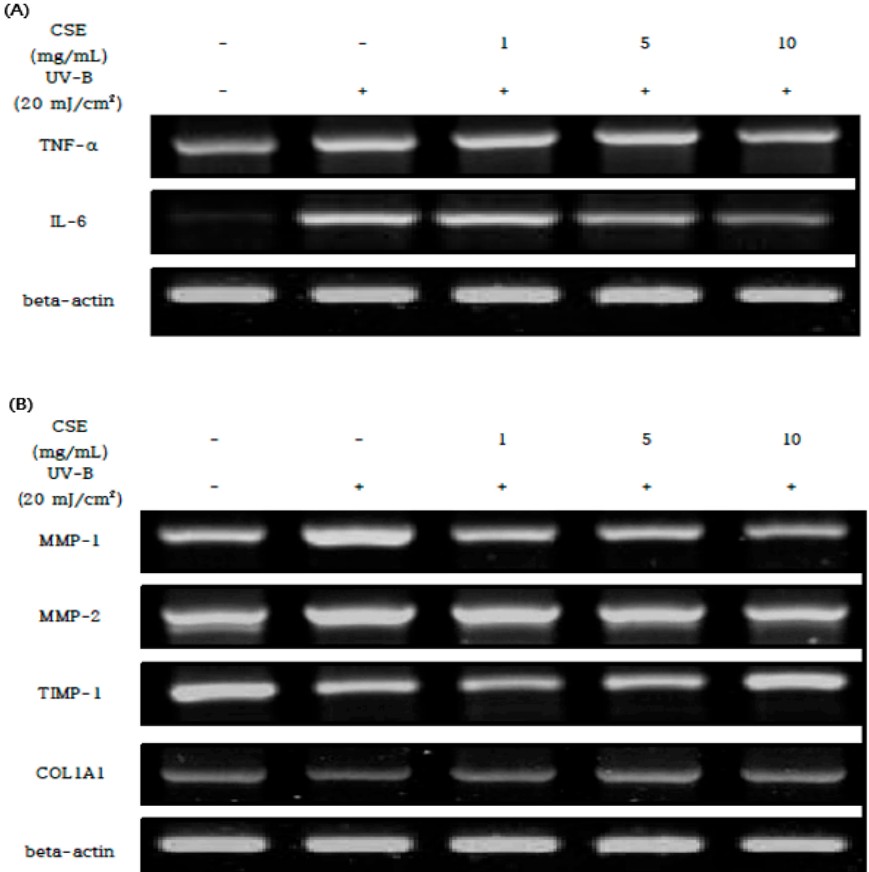

**Figure 5.** Effects of the CSE on the expression of inflammatory and photo-aging in HDFs cells. RT-PCR analysis of mRNA expression levels of inflammatory (**A**) and photo-aging (**B**). Total RNA was isolated from the HDFs cells of the normal, UV-B, and CSE (1, 5 and 10 mg/mL) groups for RT-PCR analysis at 24 h. Total RNA was isolated and quantified by beta-actin for quantitative comparison.

## 4. Discussion

Photo-aging and wrinkles are caused by the production of reactive oxygen species in skin tissue exposed to UV light and by the imbalance between MMP/TIMP in activated fibroblasts, which weakens the dermal layer's ability to regenerate cells and the factors that control skin elasticity.

The epidermis generates ROS when it is exposed to external stimuli, and the generated ROS stimulates keratinocytes to produce and secrete interleukins, such as IL-1α, IL-1β, IL-3, IL-6, and IL-8, as well as colony-stimulating factors and cytokines, such as TNF-α (tumor necrosis factor—alpha) [28,30,34,35]. When the cytokines produced and secreted by keratinocytes affect the dermal tissue in an autocrine or paracrine manner, the macrophages in the dermis react and cause complex inflammatory and immune responses or bind to each cytokine receptor in the fibroblasts to activate systems of matrix metalloproteinases (MMPs) and tissue inhibitors of metalloproteinases (TIMPs). For this reason, balancing MMPs and TIMPs is one of the most important mechanisms for improving skin diseases and inhibiting photo-aging [12,23,36].

This study evaluated the in vitro efficacy of CSE on skin cells by linking the biological expression levels of MMPs and TIMPs with skin diseases, such as wrinkles and photo-aging, in HDF cells that induced or did not induce skin diseases using UVB.

Prior to the experiment, various types of amino acids among the main components contained in CSE were analyzed through amino acid composition analysis. Among them, the presence or absence of collagen components in CSE was determined by detecting glycine, proline, and hydroxyproline constituting collagen, CSE contains various kinds of

amino acids and contains a high content of glycine (3905 μg/mL), proline (1432 μg/mL), and hydroxyproline (1554 mg/mL). These results showed that CSE contains a large amount of collagen, thus it was expected that the collagen component contained in a large amount would have a positive effect on the skin, and it was used as a basic research result to proceed with the follow-up study.

Through measuring CSE cell viability using MTT assay in HDF cells, it was determined that there was no toxicity to the cells at a concentration of 12.5 mg/mL or less, thus the subsequent tests were performed by setting the concentration to this level.

Collagen protects the skin from external stimuli or force by giving it strength and tension, and a decrease in collagen is closely related to skin aging, thus the synthesis and degradation of ECM extracellular matrix, such as collagen, needs to be controlled properly. However, excessive exposure to stimuli, such as UV rays, promotes the expression of MMPs, which are proteolytic enzymes that degrade proteins such as extracellular matrix (ECM) and basement membrane (BM), and the activated MMPs cause various skin diseases by disrupting skin tissue balance through breaking down collagen in the dermal layer. This study investigated the changes in the physiological activity of skin cells according to CES concentration (0.2 mg/mL, 0.39 mg/mL, 0.78 mg/mL, 1.56 mg/mL, 3.13 mg/mL, and 6.25 mg/mL) in HDF cell lines through MMP-1, elastase, and PIP ELISA tests. CES inhibited MMP-1 in a concentration-dependent manner by 21.89% compared to the solvent control at a concentration of 6.25 mg/mL. In the elastase inhibition assay test, CSE inhibited elastase activity in a concentration-dependent manner by 12.04% compared to the solvent control at a concentration of 6.25 mg/mL. These results suggest that CSE could prevent skin tissue decomposition and maintain and improve skin tissue conditions, and the PIP test results showed that CSE enhanced the synthesis of collagen by 62.24% at a concentration of 6.25 mg/mL by inhibiting ECM degradation.

In an experiment using RT-PCR, UV-B roll was irradiated to HDF cells to measure the efficacy of CSE treated with different concentrations (1, 5, and 10 mg/mL) on inflammation and photo-aging. CSE dose-dependently decreased UV-B-enhanced inflammatory cytokines TNF-$\alpha$ and IL-6 in skin cells. In addition, CSE regulated the mRNA expression of MMP-1, MMP-2, and TIMP-1 in the imbalance of MMPs and TIMPs caused by cytokines increased by the inflammatory response and increased the production of type I$\alpha$1 pro-collagen.

This result showed that CES could maintain and improve the condition of normal skin tissue in HDF cell lines not induced by UV-B. Furthermore, the results of inhibiting ECM degradation, such as elastin and collagen, through MMP/TIMP balancing by inhibiting and promoting MMPs and TIMPs in HDFs cell lines, where skin diseases such as photo-aging and wrinkles were induced by UV-B, confirmed the potential of CSE as a functional material that can play a significant role in skin health by inhibiting photo-aging and treating skin diseases in skin tissues damaged or undamaged by UV-B.

## 5. Conclusions

Today, fish are used not only as a variety of food but also as a health-promoting food. However, this industrial use is still limited. In particular, it is known that the nutrient components, including collagen, are distributed in the fish skin, but the development of a product using the fish skin in the industry has not been activated.

In this study, it was confirmed that CSE treatment reduced TNF-$\alpha$ and IL-6 in a dose-dependent manner after UV-B irradiation on HDF cells induced a strong inflammatory response and photo-aging. In addition, CSE inhibited the activity of MMP-1 and 2 related to skin diseases such as wrinkles and photo-aging, and increased the activity of TIMP-1, thereby inhibiting elastin degradation and increasing COL1A1 activity through the balance between MMP and TIMP.

Through these results, we would like to suggest that commercially valuable substances can be obtained using fish skins discarded as waste, and in particular, they can be used as beneficial materials for skincare. Further studies will be designed to evaluate the efficacy

as a functional material through the separation and purification of *Cynoglossus semilaevis* shell peptide components.

**Author Contributions:** Conceptualization, S.-C.C. and I.-A.L.; methodology, S.-C.C. and I.-A.L.; software, S.-C.C. and I.-A.L.; validation, S.-C.C. and I.-A.L.; formal analysis, S.-C.C. and I.-A.L.; investigation, S.-C.C. and I.-A.L.; resources, S.-C.C. and I.-A.L.; data curation, S.-C.C. and I.-A.L.; writing—original draft preparation, S.-C.C. and I.-A.L.; writing—review and editing, S.-C.C. and I.-A.L.; visualization, S.-C.C. and I.-A.L. All authors have read and agreed to the published version of the manuscript.

**Acknowledgments:** This research was supported by research funds of Kunsan National University(18B18843131) and National Research Foundation (NRF-2019R1A4A1026423).

**Conflicts of Interest:** The authors declare no conflict of interest.

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
