# Peer review of "Effect of MMP/TIMP Balancing of Cynoglossus semilaevis Shell Extracts on Skin Protection"

_fishes, doi:10.3390/fishes6030034_

Round 1

Reviewer 1 Report

I read the paper entitled:"Effect of MMP/TIMP Balancing of Cynoglossus semilaevis Shell Extracts on Skin Protection" and found it an interesting contribution.  However, several adjustments should be performed, as explain next.

Minor observations:

  1. abbreviations within the abstract should be avoided: it means to write human dermal fibroblasts instead of HDF.  Then, after defined within the manuscript it would be fine to use it. 
  2. some inconsistencies in plural/singular:
    1. lane 34 protein ....proteins
    2. lane 198 skin cell....skin cells
  3. latin words in italics: in vivo.
  4. typos:
    1. lane 89 gl2 ycine for glycine
    2. lane 115 colla genase for collagenase
    3. lane 133 collage orase
  5. paragraphs difficult to understand, suggest rephrasing with a native speaker:
    1. lanes 90-93.  Why associating aminoacids with effects on skin
    2. lanes 108-118 rephrase
    3. lanes 129-130 rephrase
    4. lanes 144-146 rephrase.

Major observations

  1. Table 1. Why the second columns is labeled as aminoacids but some of the molecules found are not.
  2. Thorugh the manuscript numbers sometimes appear with commas and others with no commas.  Follow editorial directions.
  3. Instead of 4,316 pg/ul it is more appropiate to use 4.3 ng/uL.
  4. What is the support, within the results, to state that MMP-1 reduces collagenase activity.  
  5. TGB1 as a positive control is not explained.
  6. Why ELISA results for collagen are expressed as PIP?

Experimental observations:

  1. It is said that SSE was used as control, however, no results were presented.
  2. What kind of standard reagents were used for HPLC analysis.
  3. If a quantitative PCR machine was used, why results were not presented in a quantitative manner. Only gels images, which make difficult to support the changes in gene expression informed.
  4. A reference for the primers used for PCR or alternatively the software used for self-design.

Author Response

We are grateful to the reviewers for their insightful comments on my paper. We have been able to incorporate changes to reflect most of the suggestions provided by the reviewers. we have corrected you pointed out within the manuscript.

Reviewer 2 Report

Dear authors!

This is a well-written paper. The environmental aspect is appreciated and adds value to the paper, which also is innovative.

I have some minor comments:
Would you mention any strengths and limitations at the end of the discussion, before the conclusion. (For instance that the study only shows results in vitro and that further tests would be needed before usefulness can be assessed.)

You have a lot of abbreviations that are not written out in full, including multiple abbreviations in the abstract. Please make sure that all abbreviations are written in full when mentioned for the first time in the abstract and main text.

The photos are of high quality and interesting/useful (for a reader like myself who works as a clinician and doesn't know much about fishes).

Table 1 is very nicely structured, highlighting the important results. However, in the pdf version, the last row appears on the next page alone. Perhaps adjust to fit?

Author Response

We are grateful to the reviewers for their insightful comments on my paper. We have been able to incorporate changes to reflect most of the suggestions provided by the reviewers. we have corrected the table picture you pointed out within the manuscript.
